# Offline Goal-Conditioned Reinforcement Learning with Projective Quasimetric Planning

**Anthony Kobanda**[1,3], **Waris Radji**[3], **Mathieu Petitbois**[1,2],
**Odalric-Ambrym Maillard**[3], **Rémy Portelas**[1]

`anthony.kobanda@ubisoft.com`

[1]**Ubisoft la Forge, Bordeaux, France**, [2]**University of Angers, France**,
[3]**Inria, Univ. Lille, CNRS, Centrale Lille, UMR 9198-CRIStAL, F-59000 Lille, France**

## Abstract

Offline Goal-Conditioned Reinforcement Learning seeks to train agents to reach specified goals from previously collected trajectories. Scaling that promises to long-horizon tasks remains challenging, notably due to compounding value-estimation errors. Principled geometric offers a potential solution to address these issues. Following this insight, we introduce **Projective Quasimetric Planning** (`ProQ`), a compositional framework that learns an asymmetric distance and then repurposes it, firstly as a repulsive energy forcing a sparse set of keypoints to uniformly spread over the learned latent space, and secondly as a structured directional cost guiding towards proximal sub-goals. In particular, `ProQ` couples this geometry with a Lagrangian out-of-distribution detector to ensure the learned keypoints stay within reachable areas. By unifying metric learning, keypoint coverage, and goal-conditioned control, our approach produces meaningful sub-goals and robustly drives long-horizon goal-reaching on diverse a navigation benchmarks.

## 1 Introduction

Humans improve knowledge and master complex skills by iteratively exploring and refining internal notions of proximity ("*How far am I from success ?"*) and progress ("*Is this action helpful ?*") (13; 26). Reinforcement learning (RL) (61) offers the computational analogue, combining trial-and-error with value function approximation to learn algorithms and agents to solve sequential decision problems, extended to high-dimensional perception and control with Deep Reinforcement Learning (DRL) (3).

To build AI agents in game productions, live exploration is costly ; however, vast logs of diverse human play exist. Offline RL addresses data-efficiency concerns by learning from a fixed set of trajectories collected a priori (38). Without the ability to query new actions, the agents must optimize their policy, eventually beyond the dataset behavior distribution (30; 52). Nevertheless, accurately estimating long-horizon value functions is notably difficult as bootstrapping errors may occur (50).

By conditioning an agent's policy on a target outcome, Goal-Conditioned RL (GCRL) repurposes the sequential decision problem as a directed reachability task (41; 25). Although techniques like Hindsight Experience Replay (HER) (2) mitigate sparse rewards and Hierarchical Policies (HP) (50; 40) ease path planning, most methods fall short when tackling complex long-horizon navigation problems, which is our main focus in our research. Recent benchmarks like D4RL (17) and OGBench (49) highlight these difficulties within diverse environments and multiple dataset distributions.

To tackle these failures, a relevant remedy to consider is to simplify the planning problem by learning a latent representation that captures reachability. Contrastive and Self-Supervised (35; 63) methods cluster *similar* states but lack calibrated distances ; Bisimulation metrics and MDP abstractions (60; 18) preserve dynamics but involve variational objectives and offer no out-of-distribution guard ;

Current landmark-based mapping methods pre-compute distances to fixed keypoints but may suffer coverage gaps (22) ; Recently, HILP (51) took an interesting geometric turn by embedding states into a Hilbert space where straight-line length, or Euclidean norm, approximates time-to-reach. However, at test time, it must scan the entire offline dataset for nearest neighbors, and its enforced symmetry overlooks the inherently directional nature of goal reachability.

Directionality matters : descending a cliff is simple ; climbing back is not. Quasimetric RL (QRL) (66; 67) embraces this asymmetry, learning a quasimetric (non symmetric distance) upper-bounding transitions to an unit cost while *pushing apart* non-successor states, and consequently approximates the minimum reach time between any states. Nonetheless, the policy learning is brittle as it *only* relies on model-based rollouts, which fail in long-horizon settings as inaccuracies accumulate.

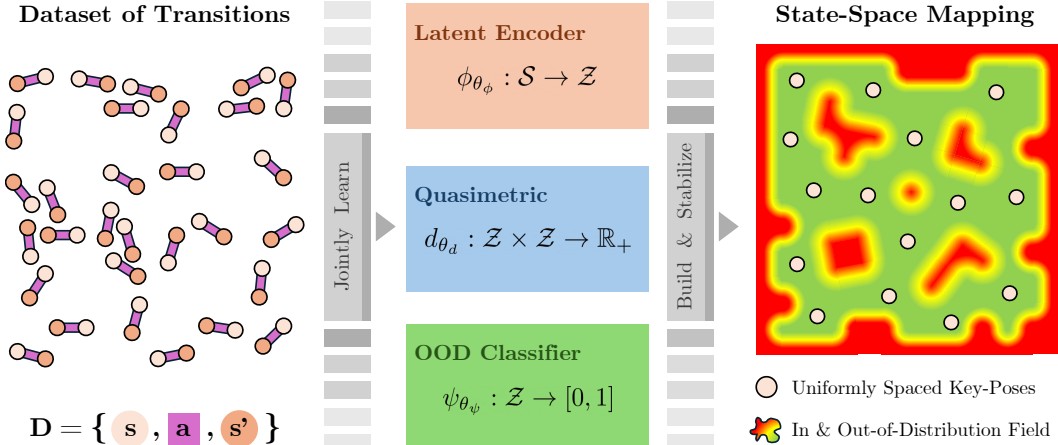

Figure 1: **Projective Quasimetric Planning (`ProQ`) : Building a Precise State-Space Mapping**. From left to right, we show how `ProQ` turns unlabeled traces into a geometry-aware navigation map: **(a)** We start with a dataset of transitions ; **(b)** We jointly train : $\phi_{\theta_\phi}$, an encoder ; $d_{\theta_d}$, a quasimetric ; $\psi_{\theta_\psi}$, an out-of-distribution classifier ; **(c)** Using $\phi$, $d$, and $\psi$, we initialize a small set of latent keypoints and let them evolve as identical particles under two *energy* based forces : a Coulomb repulsion ensuring they uniformly spread across the latent space ; an OOD barrier keeping them within the in-distribution manifold ; **(d)** To navigate the resulting space, we do path planning using Floyd-Warshall and action selection with an AWR-trained policy.

To address long-horizon navigation, we present **Projective Quasimetric Planning**, or **`ProQ`**, a physics-inspired mapping framework that converts an offline dataset into a finite and uniformly covering set of safe keypoints (see *Figure* 1). The latent geometry is shaped in a model-free way by two energy functions : a Coulomb term that pushes keypoints apart and an entropic barrier that keeps them in-distribution. Since the state encoder, the quasimetric, and the barrier (OOD discriminator) are learned simultaneously, they regularizes the others, yielding a smooth, well-covered manifold.

We use a goal-conditioned policy trained with Advantage Weighted Regression (AWR) (52) to generate actions toward nearby latent targets, a simpler learning task than direct long-range navigation. For distant goals, we employ efficient Floyd-Warshall lookups across keypoints. This approach avoids distance symmetry constraints of HILP and scales memory independently of dataset size.

Our main contributions are :

- **A unified latent geometry :** As we jointly learn a state-space encoder, a quasimetric, and an out-of-distribution classifier, the resulting space is bounded, directional, and safely calibrated.

- **A physics-driven latent coverage :** Coulomb repulsion uniformly and maximally spreads keypoints as identical particles, while a continuously repulsive OOD barrier keeps them in-distribution.

- **A graph-based navigation pipeline :** Planning reduces to simple greedy hops between keypoints, making long-horizon navigation solvable with short-horizon *optimally* selected sub-goals.

While *Section* 2 reviews the relevant and related literature, *Section* 3 presents the necessary theoretical background contextualizing our research. In *Section* 4, we detail our approach, going through the learning process and the inference pipeline. *Section* 5 presents our experimental protocol and results. We conclude in *Section* 6, notably with the limitations of our work and the next stages considered.

## 2 Related Work

**Offline Goal-Conditioned Reinforcement Learning (GCRL)**    As the name implies, Offline GCRL lies at the intersection of Offline RL (34; 38) and GCRL (25; 41), and aims to address two main challenges : trajectory stitching and credit assignment (17; 49). Prior works tackled those issues by leveraging relabeling methods. For instance, (19; 73) apply Hindsight Experiment Replay (HER) (2) with Imitation Learning. Leveraging this basis, (10; 30) additionally performs value learning and policy extraction through Offline Q-Learning. Nevertheless, both kinds of methods struggle in long-range scenarios, notably when the value estimates suffer from a low *signal-to-noise* ratio.

**Hierarchical and Graph Methods for Long-Horizon Planning**    Decomposing the decision-making by leveraging a hierarchical structure is a long-standing idea (57; 62). Recent work showed its interest in solving long-horizon tasks in online settings (64; 9; 32; 39; 44; 45; 75) and in offline ones (50; 8). While many Hierarchical GCRL (HGCRL) methods relate to the offline skill extraction literature (1; 24; 31; 53; 55; 59; 71; 50; 51; 29), which aims to decompose the goal-reaching task into a succession of sub-tasks (as trajectory segments, directions, or precise subgoals), our method relates more to the graph planning strategies. Indeed, learning graphs for planning has been broadly used in the online setting to alleviate exploration problems (14; 20; 23; 28; 27; 48; 56; 74; 21; 37; 36), unlike `ProQ`, which targets the offline setting. Recent work started to use graph learning as a way to ease long-range planning in the offline settings (76; 8), however without relying on a quasimetric distance, which allow in our framework the learning of a uniformly spanning set of in-distribution keypoints.

**Representation and Latent Metric Learning**    Leveraging temporally meaningful representations for planning in a complex state space has been a growing research topic in GCRL. While several prior works relied on contrastive losses to encode temporal similarity within the representations (15; 8; 72; 16; 76; 46), they did so without a real metric calibration, as `ProQ` does. Instead of a contrastive approach, our method leverages a distance function within a latent state space to allow for metric-grounded planning. Likewise, Foundation Policies with Hilbert Representations (HILP) (51) builds a Hilbert representation space to encode temporal similarity of states within its induced metric, but without taking into account asymmetry like our framework does. Additionally, Quasimetric Reinforcement Learning (QRL) (67), which is closely related to our proposed method, leverages Interval Quasimetric Embeddings (IQE) (66) to learn an asymmetric distance-value function. However, while QRL uses it to extract a policy through a valued model-based approach, we leverage it to guide the learning of a uniformly distributed set of keypoints to perform long-range planning.

## 3 Preliminaries

We consider a **Markov Decision Process (MDP)** as tuple $\mathcal{M} = \left( \mathcal{S}, \mathcal{A}, \mathcal{P}_{\mathcal{S}}, \mathcal{P}_{\mathcal{S}}^{(0)}, \mathcal{R}, \gamma \right)$, which provides a formal framework for RL, where $\mathcal{S}$ is a state space, $\mathcal{A}$ an action space, $\mathcal{P}_{\mathcal{S}} : \mathcal{S} \times \mathcal{A} \rightarrow \Delta(\mathcal{S})$ a transition function, $\mathcal{P}_{\mathcal{S}}^{(0)} \in \Delta(S)$ an initial distribution over the states, $\mathcal{R} : \mathcal{S} \times \mathcal{A} \times \mathcal{S} \rightarrow \mathbb{R}$ a deterministic reward function, and $\gamma \in ]0, 1]$ a discount factor. An agent's behavior follows a policy $\pi_\theta : \mathcal{S} \rightarrow \Delta(\mathcal{A})$, parameterized by $\theta \in \Theta$. In Reinforcement Learning, we aim to learn optimal parameters $\theta_{\mathcal{M}}^*$ maximizing the expected cumulative reward $J_{\mathcal{M}}(\theta)$ or the success rate $\sigma_{\mathcal{M}}(\theta)$.

**Offline Goal-Conditioned RL (GCRL)** extends the MDP to include a goal space $\mathcal{G}$, introducing $\mathcal{P}_{\mathcal{S},\mathcal{G}}^{(0)}$ an initial state and goal distribution, $\phi : \mathcal{S} \rightarrow \mathcal{G}$ a function mapping each state to the goal it represents, and $d : \mathcal{G} \times \mathcal{G} \rightarrow \mathbb{R}^+$ a distance metric on $\mathcal{G}$. The policy $\pi_\theta : \mathcal{S} \times \mathcal{G} \rightarrow \Delta(\mathcal{A})$ and the reward function $\mathcal{R} : \mathcal{S} \times \mathcal{A} \times \mathcal{S} \times \mathcal{G} \rightarrow \mathbb{R}$ are now conditioned on a goal $g \in \mathcal{G}$. In our experimental setup, we consider sparse rewards allocated when the agent reaches the goal within a range $0 \leq \epsilon$ : $\mathcal{R}(s_t, a_t, s_{t+1}, g) = \mathbb{1}\left( d(\phi(s_{t+1}), g) \leq \epsilon \right)$. Given a dataset $\mathcal{D} = \left\{ (s, a, r, s', g) \right\}$, the policy loss is optimized to reach the specified goals. This formulation represents the core problem we address.

Unlike traditional metrics, which are symmetric, an optimal value function in GCRL often exhibits asymmetry as an optimal path from a state A to a state B is may not be the same to go from B to A. This property aligns with the concept of a **Quasimetric** (70), a function $d : \mathcal{G} \times \mathcal{G} \rightarrow \mathbb{R}^+$ satisfying three rules : *Non-negativity* : $d(x, y) \geq 0$ ; *Identity of indiscernibles* : $d(x, x) = 0$ ; *Triangle inequality* : $d(x, z) \leq d(x, y) + d(y, z)$ ; but not necessarily symmetry : $d(x, y) \neq d(y, x)$.

To model such a distance, we consider : **Interval Quasimetric Embeddings (IQE)** (66), and its continuation, **Quasimetric Reinforcement Learning (QRL)** (67). IQE represents the quasimetric as the aggregated length of intervals in a latent space and is a universal approximation for distances.

IQE considers input latents as two-dimensional matrices (via reshaping). Given an encoder function from state space $\mathcal{S}$ to a latent space $\mathbb{R}^{N \times M}$ $f_{\theta'} : \mathcal{S} \to \mathbb{R}^{N \times M}$, the distance from $x \in \mathcal{S}$ to $y \in \mathcal{S}$ is formed by components that capture the total size (i.e., Lebesgue measure) of unions of several intervals on the real line :

$$\forall\, i = 1, \ldots, N \qquad d_{\theta'}^{(i)}(x, y) = \left| \bigcup_{j=1}^{M} \Big[ f_{\theta'}(x)_{ij}, \max\big(f_{\theta'}(x)_{ij}, f_{\theta'}(y)_{ij}\big) \Big] \right| \tag{1}$$

IQE components are positive homogeneous and can be arbitrarily scaled, and thus do not require special reparametrization in combining them. Using the *maxmean* reduction from prior work (54), we obtain IQE-*maxmean* with a single extra trainable parameter $\alpha \in [0, 1]$, and considering $\theta = [\theta', \alpha]$ :

$$d_\theta(x, y) = \alpha \cdot \max\big(d_{\theta'}^{(1)}(x, y), .., d_{\theta'}^{(N)}(x, y)\big) + (1 - \alpha) \cdot \text{mean}\big(d_{\theta'}^{(1)}(x, y), .., d_{\theta'}^{(N)}(x, y)\big) \tag{2}$$

**Theorem 3.1.** *IQE Universal Approximation General Case (See (66) for the demonstration) : Consider any quasimetric space $(\mathcal{X}, d)$ where $\mathcal{X}$ is compact and $d$ is continuous. $\forall\, \epsilon > 0$,- with sufficiently large $N$, there exists a continuous encoder $f_{\theta'} : \mathcal{X} \to \mathbb{R}^{N \times M}$ and $\alpha \in [0, 1]$ such that :*

$$\forall x, y \in \mathcal{X}, \; \big|d_\theta(x, y) - d(x, y)\big| \leq \epsilon .$$

IQE formulation captures directionality and satisfies the quasimetric properties, making it suitable for modeling asymmetric relationships in GCRL. QRL focuses on learning a quasimetric from data that aligns with the optimal value function, with the learning objective over an IQE function enforcing : *Local Consistency*, as for an observed transition $(s, a, s')$ $d(s, s') = 1$ ; and *Global Separation*, as for non-successor pairs $(s, s'')$ it ensures $d(s, s'')$ is maximal. The loss function combines these aspects :

$$\mathcal{L}_{QRL}(\theta) = \max_{\lambda \geq 0} \mathbb{E}_{(s, s', s'') \sim \mathcal{D}} \Big[ \lambda \cdot \big(\texttt{relu}(d_\theta(s, s') - 1)^2 - \epsilon^2\big) - \omega\big(d_\theta(s, s'')\big) \Big], \tag{3}$$

where $\epsilon$ relaxes the local constraint and consequently eases the training, while $\omega$ is a monotonically increasing convex function to speed the convergence and stabilize the learning process.

**Theorem 3.2.** *QRL Function Approximation [General,Formal] (See (67) for the demonstration) : Consider a compact state space $\mathcal{S}$ and an optimal value function $V^*$. We suppose that the state space is the goal space ($\mathcal{S} = \mathcal{G}$). If $\{d_\theta\}_{\theta \in \Theta}$ are universal approximators of quasimetrics over $\mathcal{S}$ (e.g. IQE) :*

$$\forall \epsilon > 0, \; \exists \theta^* \in \Theta, \forall s, g \in \mathcal{S}, \; \mathbb{P}\Big[\big|d_{\theta^*}(s, g) + (1 + \epsilon)V^*(s, g)\big| \in \big[-\sqrt{\epsilon}, 0\big]\Big] = 1 - \mathcal{O}\big(-\sqrt{\epsilon} \cdot \mathbb{E}[V^*]\big)$$

*i.e. $d_{\theta^*}$ recovers $-V^*$ up to a known scale, with probability $1 - \mathcal{O}\big(-\sqrt{\epsilon} \cdot \mathbb{E}[V^*]\big)$ .*

Since one of our objectives is to map the goal space $\mathcal{G}$ with a set of keypoints $\{z_k\}_{k=1}^K$, to ensure comprehensive coverage, we draw inspiration from physics, particularly the repulsive forces observed in charged particles. In electrostatics, according to Coulomb's law, two popositively charged particles located at $x_1, x_2 \in \mathbb{R}^3$ experience a potential :

$$V(x_1, x_2) = \frac{q_1 q_2}{4\pi\epsilon_0} \cdot \frac{1}{\|x_1 - x_2\|_2},$$

where $q_1, q_1 \in \mathbb{R}$ are their charges, $\epsilon_0$ is the vacuum permittivity, and $\|\cdot\|_2$ is the Euclidean norm. For K charges, when all charges are equal ($q_i = q_j$), we often only study the total potential energy :

$$E(x_1, \ldots, x_K) = \sum_{1 \leq i < j \leq K} \frac{1}{\|x_i - x_j\|_2} .$$

Minimizing this energy $E(x_1, \ldots, x_K)$ on a compact domain (e.g. the unit sphere $\mathbb{S}^2$) yields the classical Thomson Problem (7; 33) of finding well-separated, nearly uniform point configurations.

Adapting this concept, we model a set of keypoints $\{z_k\}_{k=1}^K$ that repel each other to achieve uniform distribution. As a disclaimer, with further explanation provided in the *Section 4*, this analogy serves as a guide ; we do not simulate physical dynamics but rather use the concept for our loss design.

## 4 Projective Quasimetric Planning

We now describe Projective Quasimetric Planning (ProQ). *Section 4.1* offers a summary of the algorithm's structure. *Section 4.2* elaborates on how ProQ learns a robust latent representation of the state space. Next, *Section 4.3* details the physics-inspired mechanism to learn keypoints within this learned latent space. Finally, *Section 4.4* describes how these keypoints are leveraged for planning.

### 4.1 ProQ Learning Algorithm : Overview

ProQ is a compositional framework that learns robust state-space representations from offline data and enables efficient goal-conditioned planning through graph-based navigation. The approach jointly trains three neural components: a state encoder $\phi$, an asymmetric quasimetric distance function $d$, and an out-of-distribution classifier $\psi$. To enable long-horizon planning, ProQ also learns a sparse set of latent keypoints that provide uniform coverage of the learned geometry, forming a navigation graph for efficient goal-reaching. The learning pipeline is presented in *Algorithm 1*, with a thorough description and explanation in *Section 4.2*. The inference pipeline is presented in *Algorithm 2*

---

**Algorithm 1 ProQ – Projective Quasimetric Planning (Learning Pipeline)**

---

**Require:** dataset $\mathcal{D}$ ; batch size $B$, nb. of keypoints $K$ ; nb. of epochs $E$ ; multipliers $\lambda_{\text{dist}}, \lambda_{\text{ood}}, \lambda_{\text{kps}}$.

1: **Init:** neural networks $\phi_{\theta_\phi}$ (latent encoder) , $\psi_{\theta_\psi}$ (OOD detector), $d_{\theta_d}$ (quasimetric), $\pi_{\theta_\pi}$ (actor) ;
2: **Init:** keypoints $\{p_k\}_{k=1}^K \subset \mathcal{D}$ ;

3: **for** $t = 1$ to $E$ **do**

4:  **Sample a batch of $B$ transitions** and random value-actor goals : $\{(s_i, a_i, s_i', g_i^v, g_i^a)\}_{i=1}^B$
5:  **Convex interpolation** (for OOD learning) : $\tilde{s}_i^c = (1 - \alpha_i)s_i + \alpha_i g_i^v$ , $\alpha_i \sim U[0, 1]$
6:  **Convex extrapolation** (for OOD learning) : $\tilde{s}_i^e = (1 + \beta_i)s_i - \beta_i g_i^v$ , $\beta_i \sim U[0, 1]$
7:  **Encode** : $s_i, s_i', g_i^v, g_i^a, \tilde{s}_i^c, \tilde{s}_i^e \xrightarrow{\phi} z_i, z_i', z_i^v, z_i^a, \tilde{z}_i^c, \tilde{z}_i^e$

8:  *(A) Latent–space learning* ( these losses are back-propagated on all $\phi, \psi, d$ ) : *Section 4.2*
9:  **Representation Loss :** $\mathcal{L}_{\text{repr}}(z_i) = \mathcal{L}_{\text{var}}(z_i) + \mathcal{L}_{\text{cov}}(z_i)$ *( Eq. 4 & 5 )*
10:  **Distance loss :** $\mathcal{L}_{\text{dist}}(z_i, z_i', z_i^v) = \lambda_{\text{dist}} \cdot \mathcal{L}_{\text{close}}(z_i, z_i') + \mathcal{L}_{\text{far}}(z_i, z_i^v)$ *( Eq. ?? & 6 )*
11:  **OOD loss :** $\mathcal{L}_{\text{ood}}(z_i, \tilde{z}_i^c, \tilde{z}_i^e) = \mathcal{L}_{\text{id}}(z_i) + \mathcal{L}_{\text{inter}}(\tilde{z}_i^c) + \mathcal{L}_{\text{extra}}(\tilde{z}_i^e)$ *( Eq. 7 )*

12:  *(B) Keypoint coverage* ( where $z_k = \phi(p_k)$ ) : *Section 4.3*
13:  **Particles loss :** $\mathcal{L}_{\text{kps}}(\{p_k\}_{k=1}^K) = \sum_{i \neq j} \mathcal{L}_{\text{repel}}(z_i, z_j) + \sum_k \mathcal{L}_{\text{ood}}(z_k)$ *( Eq. 8 & 20 )*

14:  *(C) Actor (AWR) loss*
15:  **Actor loss :** $\mathcal{L}_{\text{awr}}(z_i, z_i', z_i^a) = Adv(z_i, z_i', z_i^a) \cdot \mathcal{L}_{\text{bc}}(z_i, a_i)$ *( Eq. 10 & 11)*

---

**Algorithm 2 PROQ – Projective Quasimetric Planning (Inference Pipeline)**

---

**Require:** initial state $s_0 \in \mathcal{S}$ , goal $g \in \mathcal{G}$ , latent keypoints $\{z_k\}_{k=1}^K$ , maximum horizon $T$ .

1: **Init:** Build $G = \{z_k\}_{k=1}^{K+1}$ with $z_{k+1} = \phi(g)$ .
2: **Init:** Compute $D^* = Distances^*\left(\{z_k\}_{k=1}^{K+1}\right) \leftarrow$ Floyd–Warshall$(G)$ ;
3: **Init:** $t \leftarrow 0$ , $s \leftarrow s_0$ ;
4: **while** $t < T$ **and** $\epsilon < d\left(\phi(s), \phi(g)\right)$ **do**
5:  $z_s \leftarrow \phi(s)$ ;
6:  $k^* = \arg\min_k \left[d_\theta(z_s, z_k) + D^*\left(z_k, \phi(g)\right)\right]$ ;
7:  $a \sim \pi_{\theta_\pi}\left(\cdot \mid s, z_{k^*}\right)$ ;
8:  $s \leftarrow$ env.step$(a)$ ;
9:  $t \leftarrow t + 1$ .

---

## 4.2 Learning a Robust Latent Space

**State-Encoder Learning (Representation Loss)** To obtain a structured latent space that captures meaningful differences between states, we encode each observation $s$ into a latent vector $z = \phi(s)$ using a neural encoder $\phi$ trained with VICReg (4). VICReg balances three terms : (1) an invariance loss aligning paired embeddings ; (2) a variance regularization ensuring non-collapsed features; and (3) a decorrelation loss to prevent redundancy. We consider the variance and covariance regularizers :

$$\mathcal{L}_{\text{var}}(X) = \frac{1}{D} \sum_{i=1}^{D} \max\left(0, \gamma - \sqrt{\text{Var}(x^i)}\right) \qquad (4) \qquad \mathcal{L}_{\text{cov}} = \frac{1}{D} \sum_{i \neq j} C(X)_{ij}^2 \qquad (5)$$

where $X = \{X_k\}_{k=1}^N$ are encoded states, $x^i$ is the vector composed of each value at dimension $i$, $C(X)$ is the covariance matrix of $X$, $D$ the latent dimension, and $\gamma$ a target variance threshold. This encourages each dimension of the representation to capture distinct, informative features in the data.

**Quasimetric Learning (Distance Loss)** Going further, we learn a latent-space distance function $d_{\theta_d}$ that approximates the goal-conditioned value function and satisfies the properties of a quasimetric. For this, we employ IQE (see *Section 3* for more precision on the IQE and QRL frameworks), and train following the QRL trategy using the loss function provided in *Equation 3*, with :

$$\omega(z_i, z_i^v) = \sigma \cdot \texttt{softplus}\left(\frac{\mu - d(z_i, z_i^v)}{\sigma}\right) \quad \text{with } \mu = 500, \ \sigma = 0.1 \qquad (6)$$

Together, these losses push the quasimetric to match temporal proximity for known transitions while maximizing separation elsewhere, and the learning of this distance helps shape the latent space.

**Out-of-Distribution Detection (OOD Loss)** To keep the learned keypoints inside reachable areas, we train an OOD classifier $\psi : \mathcal{Z} \to [0, 1]$. The difficulty is that the dataset only supplies *positive* (in-distribution) examples. We therefore generate *negative* samples under the mild assumption that the navigation space is closed and bounded, and any state obtained by interpolation (convex hull) or extrapolation (extended convex hull) should be unreachable, unless it matches a positive state.

- **In distribution states :** $z = \phi(s)$, with the target $\psi(z) = 1$ ;
- **Interpolated states :** $\tilde{z}^c = \phi\big((1 - \alpha) \cdot s_1 + \alpha \cdot s_2\big), \alpha \sim \mathcal{U}([0, 1])$ ;
- **Extrapolated states :** $\tilde{z}^e = \phi\big((1 + \beta) \cdot s_1 - \beta \cdot s_2\big), \beta \sim \mathcal{U}([0, 1])$ .

Using these generated states, we learn $\psi$ with a binary cross-entropy with a Lagrange multiplier to enforce high confidence on positives while pushing negatives in-distribution probability towards 0 :

$$\mathcal{L}_{\text{ood}}(z_i, \tilde{z}_i^c, \tilde{z}_i^e) = -\lambda_{\text{ood}} \log \psi(z_i) - \log\big(1 - \psi(\tilde{z}_i^c)\big) - \log\big(1 - \psi(\tilde{z}_i^e)\big) , \qquad (7)$$

with $\lambda_{\text{ood}}$ updated until satisfaction of a penalty $\big((1 - \delta) - \psi(z)\big)^2$, with $0 < \delta < 1$ the compliance.

**Theorem 4.1.** *In/Out-of-Distribution Generalization Guarantee (See Appendix D.1.1 for proof) : Let $\mathcal{D}$ be an $\epsilon$-dense dataset of points within a compact state space $\mathcal{S} \subset \mathbb{R}^n$. We consider $\widetilde{\mathcal{D}}$, a set of generated points from $\mathcal{D}$, which is $\epsilon$-dense in the shell $\big\{ \tilde{s} \in \mathbb{R}^n \mid dist(\tilde{s}, \mathcal{D}) \leq Diam(\mathcal{D}) \big\}$. After convergence of the latent space component $\phi$, $\psi$, and $d$, considering the compliance $0 < \delta < 1$, there exist constants $0 < C, 0 < \eta < \text{Diam}(\mathcal{D}), 0 < \bar{\delta} < 1$ such that:*

*(ID)* $\quad \forall s \in \mathcal{S} : \qquad\qquad\qquad\qquad\qquad \psi\big(\phi(s)\big) \quad \geq \quad 1 - \delta - C \cdot \epsilon$

*(OD)* $\quad \forall x \in \mathbb{R}^n \ s.t. \ \eta \leq dist(x, \mathcal{D}) \leq Diam(\mathcal{D}) : \ \psi\big(\phi(x)\big) \quad \leq \quad \bar{\delta} + C \cdot \epsilon$

Hence, *Theorem 4.1* guarantees that, after training, the OOD classifier $\psi$ remains uniformly bounded away from ambiguity both inside the reachable region and in the controlled negative shell. In practice, this yields a well-calibrated decision boundary over latent space that reliably guides keypoint placement and downstream planning (see *Appendices D.1.1* and *D.1.2*).

## 4.3 Uniformly Mapping the Latent Space

The utility of a learned latent space for planning is significantly enhanced by having sparse, uniformly distributed *keypoints*, serving as landmarks in the learned geometry. To achieve this uniform coverage, we draw inspiration from physics, specifically the repulsive forces between like-charged particles.

**Coulomb-like Repulsion :** For latent keypoints $\{z_k\}_{k=1}^K$ we minimize the repulsion energy :

$$\mathcal{L}_{\text{repel}} = \lambda_{\text{repel}} \sum_{i \neq j} \frac{1}{d_\theta(z_i, z_j) + \epsilon}, \tag{8}$$

where $d_\theta$ is the learned quasimetric, and the $\epsilon$ a term to avoid numerical blow-up. The form mirrors Coulomb's energy, which falls off with distance and drives points towards a uniform layout.

**Entropic OOD Barrier :** Repulsion alone could push keypoints outside the region covered by the dataset. We therefore add a soft barrier derived from the OOD classifier $\psi$, penalizing points that leave the in-distribution set :

$$\mathcal{L}_{\text{ood}} = -\lambda_{\text{ood}} \cdot \sum_{k=1}^K \log \psi(z_k), \tag{9}$$

with $\lambda_{\text{ood}}$ updated until satisfaction of a penalty $\left((1 - \delta) - \sum_{k=1}^K \psi(z_k)\right)^2$, with $0 < \delta < 1$ the compliance. In other words, on average, all latent keypoints must stay in-distribution to some extent.

The energy-based losses add up to $\mathcal{L}_{\text{kps}} = \mathcal{L}_{\text{rep}} + \mathcal{L}_{\text{ood\_kps}}$. At each gradient step, while optimizing this composed loss, $\phi$, $d$, and $\psi$ are frozen to avoid instabilities in the learning of the latent space.

**Theorem 4.2.** *Existence and Distinctness of Coulomb Energy Minimizers (See Appendix D.2.2) :*
*Let's consider a repulsive energy function $E : \mathcal{S}^K \to \mathbb{R}^+ \cup \{+\infty\}$ , $\{s_k\}_{k=1}^K \to \sum_{i \neq j} \frac{1}{d(s_i, s_j)}$.*
*Under the assumptions of compactness of the state space $\mathcal{S}$ and continuity the quasimetric $d$ we have :*

*(i) $E$ has a finite minimum, and there exists a configuration $\{s_k^*\}_{k=1}^K \subset \mathcal{S}$ that reaches it.*
*(ii) For any such minimizer $\{s_k^*\}_{k=1}^K \subset \mathcal{S}$ , if $K > 1$ then all keypoints are distinct.*

## 4.4 Graph-based Navigation

With the latent geometry shaped and the keypoints frozen, inference becomes a two–stage process :
*(i)* a discrete planner selects an intermediate key point that minimises the estimated time-to-goal, and
*(ii)* a continuous controller (AWR policy) produces actions to move the agent towards that keypoint.

**From Keypoints to a Directed Graph :** Let's consider $\mathcal{V}(g) = \{z_1, \ldots, z_K, z_g\}$ be the latent key-points plus the current goal embedding. We keep an edge $(i, j) \in \mathcal{E}(g)$ only if $d(z_i, z_j) \leq \tau$ and set its weight to that distance. Here, $\tau$ is a stabilizing cut distance to avoid using potentially ill-approximated long distances. Consequently, we produce a directed graph $G(g) = (\mathcal{V}(g), \mathcal{E}(g))$, and all-pairs shortest paths are obtained with Floyd–Warshall algorithm (69), yielding a matrix $D(g)^*$ that stores the distances from every keypoint to the goal within the learned graph.

**Learning a Controller for Action Generation :** From a dataset transition $(s_i, a_i, s_i')$ we sample a relatively short-range goal $g$ and compute the distance improvement $A_i$ in *Equation 10*. Advantage-Weighted Regression (52) then fits a policy $\pi_{\theta_\pi}$ with the log-likelihood loss (11), weighted by $\exp(\alpha \cdot A_i)$. We thus obtain a robust controller used to go from one keypoint to another.

$$A_i = d\big(\phi(s_i), \phi(g_i)\big) - d\big(\phi(s_i'), \phi(g_i)\big) , \tag{10}$$

$$\mathcal{L}_{\text{awr}} = -\mathbb{E}_{(s_i, a_i)} \big[\exp(\alpha \cdot A_i) \cdot \log \pi_{\theta_\pi}(a_i \mid s_i, g_i)\big] , \tag{11}$$

This regression objective has proven stable and effective for short-horizon offline control (30; 50; 8).

# 5 Experiments

## 5.1 Environments & Datasets

We evaluate on the OGBench suite (49), a recent benchmark tailored to Offline GCRL. We focus on the POINTMAZE environments, where a point-mass must reach goal locations across mazes of different scales (*Figure 2*) and a harder teleport variant with stochastic non-local transitions.

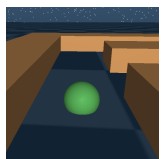 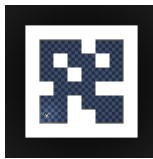 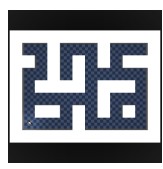 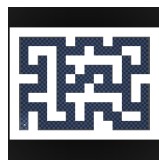 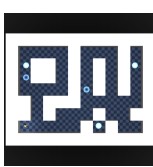

(a) Point Agent.  (b) Medium.  (c) Large.  (d) Giant.  (e) Teleport.

Figure 2: Layouts of the four POINTMAZE tasks used in our experiments. All images are rendered on the same grid resolution (blue squares have identical side length), so the overall arena grows from *Medium* to *Giant*. The *Teleport* task adds blue portal cells that stochastically move the agent to another non-local location.

OGBench provides two types of $1M$ transition datasets : *navigate*, containing demonstrations from an expert going to randomly and regularly sampled locations along long trajectories (1000 steps) ; *stitch*, containing short trajectories (200 steps) created similarly to the *navigate* ones.

We compare our proposed method to six baselines : Goal-Conditioned Behavioral Cloning (GCBC) (12), GC Implicit Value Learning (GCIVL) (50), GC Implicit Q Learning (GCIQL) (30), Contrastive RL (CRL) (15), Quasimetric RL (QRL) (67), and Hierarchical Implicit Q Learning (HIQL) (50).

## 5.2 Results

### 5.2.1 Performances

Table 1: Success rates (%) over POINTMAZE tasks. Bold entries mark the top performer in each row. `ProQ` performs the best in almost all cases, demonstrating its strong coverage and planning across different settings.

| Environment | Dataset | GCBC | GCIVL | GCIQL | QRL | CRL | HIQL | PROQ |
|---|---|---|---|---|---|---|---|---|
| pointmaze | pointmaze-medium-navigate-v0 | $9 \pm 6$ | $63 \pm 6$ | $53 \pm 8$ | $82 \pm 5$ | $29 \pm 7$ | $79 \pm 5$ | $\mathbf{100} \pm 0$ |
| | pointmaze-large-navigate-v0 | $29 \pm 6$ | $45 \pm 5$ | $34 \pm 3$ | $86 \pm 9$ | $39 \pm 7$ | $58 \pm 5$ | $\mathbf{99} \pm 1$ |
| | pointmaze-giant-navigate-v0 | $1 \pm 2$ | $0 \pm 0$ | $0 \pm 0$ | $68 \pm 7$ | $27 \pm 10$ | $46 \pm 9$ | $\mathbf{92} \pm 3$ |
| | pointmaze-teleport-navigate-v0 | $25 \pm 3$ | $\mathbf{45} \pm 3$ | $24 \pm 7$ | $4 \pm 4$ | $24 \pm 6$ | $18 \pm 4$ | $43 \pm 0$ |
| | pointmaze-medium-stitch-v0 | $23 \pm 18$ | $70 \pm 14$ | $21 \pm 9$ | $80 \pm 12$ | $0 \pm 1$ | $74 \pm 6$ | $\mathbf{99} \pm 1$ |
| | pointmaze-large-stitch-v0 | $7 \pm 5$ | $12 \pm 6$ | $31 \pm 2$ | $84 \pm 15$ | $0 \pm 0$ | $13 \pm 6$ | $\mathbf{99} \pm 1$ |
| | pointmaze-giant-stitch-v0 | $0 \pm 0$ | $0 \pm 0$ | $0 \pm 0$ | $50 \pm 8$ | $0 \pm 0$ | $0 \pm 0$ | $\mathbf{99} \pm 1$ |
| | pointmaze-teleport-stitch-v0 | $31 \pm 9$ | $\mathbf{44} \pm 2$ | $25 \pm 3$ | $9 \pm 5$ | $4 \pm 3$ | $34 \pm 4$ | $35 \pm 7$ |

Across all maze sizes and dataset types, `ProQ` consistently outperforms prior methods-particularly on the large and giant mazes. Its $92\%$ score on `giant-navigate` (versus $99\%$ score on `giant-stitch`) suggests that training on very long expert traces can introduce low-level control noise when learning. In the stochastic `teleport` variants `ProQ`'s is slightly below the best score. Moreover, `ProQ`'s lower standard deviations reflect a more stable learning pipeline.

### 5.2.2 Learned Mapping

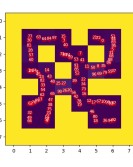 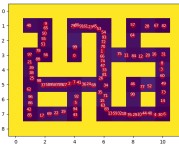 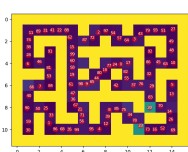 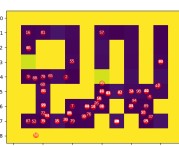

(a) Medium.  (b) Large.  (c) Giant.  (d) Teleport.

Figure 3: Illustration of the learned latent mappings produced by `ProQ` on the four POINTMAZE tasks. **OOD** probabilities are shown from $0 \sim$ *yellow* to $1 \sim$ *magenta* ; The learned keypoints are represented in red.

**OOD Mapping :** *Figure 3* shows that the classifier $\psi$ sharply separates traversable corridors (magenta) from walls and the exterior (yellow) across all mazes. The boundary follows the wall geometry, confirming that the interpolation/extrapolation training method gives a tight approximation of the reachable set. In the `teleport` variants, isolated green tiles appear. They correspond to portal entrance cells, which are therefore unstable, illustrating that $\psi$ accommodates non-trivial transitions.

**Keypoints Placement :** The Coulomb repulsion seems to properly spread the keypoints (red dots in *Figure 3*) almost uniformly along every corridor, while the OOD barrier keeps them centered. Coverage remains dense even in the wide `Giant` maze, confirming size-invariance of the mechanism. For `teleport`, we observe that in sparsely covered hallways, the equilibrium seems harder to attain. This illustrates a current limitation of `ProQ` : when the environment features stochastic transitions, the quasimetric and the force field struggle to allocate landmarks evenly, reducing coverage quality.

### 5.2.3 Path Planning

*Figure 4* depicts two sets of keypoints selected for different start and goal locations. The chosen keypoints trace the intuitive shortest corridors and never cut through walls, indicating that the learned quasimetric yields accurate one-step costs and that the offline Floyd–Warshall search exploits them effectively. Even on the longest route, only a few hops are required, demonstrating how a *sparse* graph can support near-optimal long-horizon navigation.

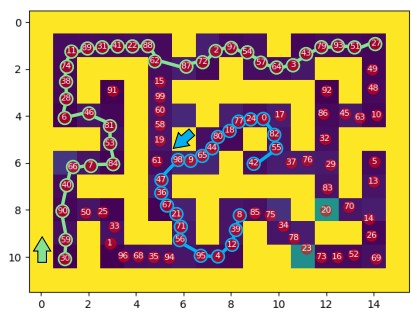

Moreover, when the goals are fixed, the all-pairs shortest-path matrices only need to be computed once, and after that, each step is a lightweight lookup over these matrices.

Figure 4: Two plans produced by `ProQ` with Floyd–Warshall on the `giant` maze.

### 5.2.4 Ablation: Why do we need an OOD detector ?

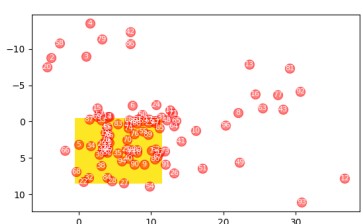

Figure 5: Key points learned without the OOD barrier on the `giant` maze; the yellow area is the reachable manifold.

Removing the OOD term leaves only Coulomb repulsion. As *Figure 5* shows, this drives many keypoints far outside the yellow support of the maze: indeed, repulsion keeps pushing points toward low-density regions where there is no counter-force. Consequently, the navigation graph no longer only covers the maze, and the quasimetric can route through "phantom" regions, underestimating travel times and yielding infeasible plans. This ablation confirms the role of the OOD barrier: it anchors the particle system to the data manifold, preserving both uniform coverage and the reliability of the learned distance.

## 6 Discussion

As takeaways, `ProQ` unifies metric learning, OOD detection, and keypoint coverage in a single latent geometry. A few hundred landmarks and one Floyd-Warshall pass per goal are enough to reach POINTMAZE targets with state-of-the-art success, showing that simple graph search, when built on a well-shaped quasimetric, can rival heavier value-based pipelines.

Nevertheless, there remain some gaps : (i) *stochastic transitions* seem to break the learning of the keypoints, as they degrade their spread ; (ii) our study is confined to POINTMAZE, hence validating on richer manipulation or vision-based tasks is required for stronger evidence ; (iii) the work is mostly empirical, and formal guarantees that we learn a satisfiable latent space component are still missing.

Future work involves various themes. First, the graph abstraction naturally supports *at inference replanning* upon environment changes. Extending this idea to lifelong *Continual* learning could unlock efficient transfer across tasks. Second, adapting the energy function so that keypoints' density grows or shrinks with local action complexity would make `ProQ` scalable to high-DOF manipulation. Finally, a tighter theoretical analysis of convergence and optimality, and broader empirical studies, will further clarify when and why quasimetric planning is preferable to value-based alternatives.

# A    Reinforcement Learning & Video Games

## A.1    Virtual Worlds, Players, and Data

Modern video games are persistent, high-fidelity simulations that run at 30 to 240 FPS and may track every interactive entity in real time, from characters and projectiles to used widgets or commands (42; 78). Worldwide platforms such as *Steam* or *Roblox* routinely serve $10^5$ to $10^7$ concurrent users, generating petabytes of play-logs per month through automatic telemetry pipelines.

These logs are archived for anti-cheat, matchmaking, and long-tail analytics. Because the underlying engines already enforce consistent physics and rendering, the recorded trajectories form an unusually clean, richly annotated dataset, in orders of magnitude larger than typical robotics corpora. Thus, this makes video games an attractive, low-risk domain for Offline Reinforcement Learning (RL) (38).

However, key challenges remain in this field of resarch : state spaces can be partially observable (fog-of-war), multi-modal (text chat, voice) and strongly stochastic (loot tables, human behavior). Furthermore, commercial constraints impose strict inference budgets and hard guarantees of player safety, pushing research toward data-efficient, controllable training settings and methods.

## A.2    From Scripted NPCs to Deep Agents

Traditional Non-Player Characters (NPC) rely on finite-state machines or behavior trees (77; 68), which are robust for systematic encounters yet brittle when mechanics change. Deep Learning (DL) entered the scene with convolutional policies for Atari (58), then self-play agents that mastered *Dota-2* (6) by running billions of simulation steps on massive clusters. While impressive, such training is impractical for most studios, due to server costs and the need for large-scale exploration.

Hybrid approaches emerged, such imitation pre-training followed by limited on-policy finetuning (47; 76), but still require intrusive engine instrumentation. Consequently, interest is shifting to *pure offline* methods to learn from existing replays to create bots or tutorial companions.

## A.3    Offline Goal-Conditioned RL in Games

Offline Goal-Conditioned RL (GCRL) (38) is a natural fit for modern games : every quest, checkpoint, or capture flag supplies an explicit goal signal, while replay archives furnish millions of feasible, human-level trajectories on which to train. Yet three practical challenges still block adoption :

**Coverage :** Even the largest replay buffers may sample only a thin manifold of the reachable world ; an agent must extrapolate reliable distances and value estimates to unexplored corners of the available maps, caves, or secret rooms.

**Directionality :** Many mechanics are inherently one-way, notably for ledge drops, conveyor belts, or one-shot teleporters. Hence, symmetric distances underestimate the true return time.

**Stochasticity :** Random critical damage, loot tables, or portal destinations may break the deterministic assumptions of classic graph planners ; consequently policies must be robust to distributional change.

## A.4    `ProQ` : Offline Goal-Conditioned RL with Navigation Graphs

Within our rsearch, our aim is to adapt the idea of *nav-meshes* (11; 5), the hand-authored graphs used in commercial engines, into data-driven learning algorithms. `ProQ` (see *Section 4*) turns trajectory datasets into a navigation graph, whose edge costs are backed by quasimetric distance weights.

Because of its structure, we infer for future work that the same agent can be fine-tuned on new maps or game modes simply by relearning the key-point set, enabling rapid transfer or continual learning. The pre-computed distances should allow fast on-line re-planning : when dynamic events reshape the layout, only the affected edges should be removed, letting the policy adapt without retraining.

# B Environment Details

## B.1 Learning Agent & Mazes

OGBench is a standardized suite of offline goal-conditioned learning tasks. It includes POINTMAZE, a simple 2D navigation environment in which a point-mass agent (with continuous $(x, y)$ position and bounded velocity inputs) must reach arbitrary goals within a bounded maze.

Four maze variants appear in OGBench (see *Figure 6*) : medium whose moderate trajectory lengths tests basic navigation ; large with longer corridor lengths increase the decision complexity ; giant scales the maze again, yielding an even larger layouts that force an agent to compose many short hops into a long-horizon path. teleport is of the same size as the large with stochastic portals that randomly chose exits (potentially a dead end).

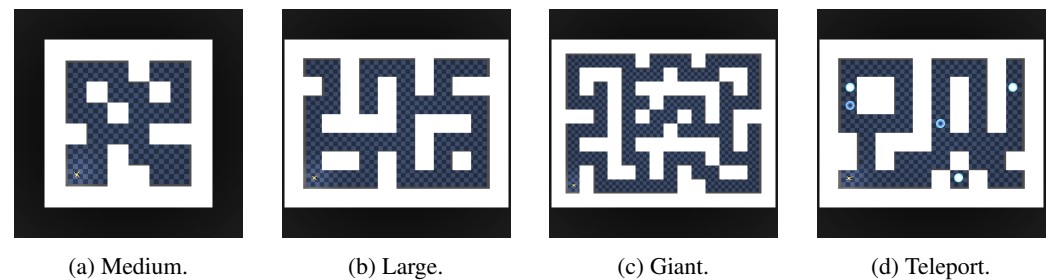

| (a) Medium. | (b) Large. | (c) Giant. | (d) Teleport. |

Figure 6: Layouts of the four POINTMAZE tasks used in our experiments. All images are rendered on the same grid resolution (blue squares have identical side length), so the overall arena grows from *Medium* to *Giant*. The *Teleport* task adds blue portal cells that stochastically move the agent to another non-local location.

## B.2 Datasets

Each layout provides two offline datasets of one million transitions. navigate contains long expert rollouts (1000 steps each) where a noisy expert wanders toward random goals ; these traces densely cover corridors but compound low-level control noise. stitch breaks those same trajectories into many shorter ones (200 steps each), forcing the agent to *stitch* discontiguous segments.

*Figure 7* show illustrations of percentages of the available trajectories for the two dataset types. We notably see how densely the trajectories cover each maze, for both navigate and stitch.

# C Implementation Details

In our experiments we use JAX as a learning framework, with FLAX and OPTAX implementation tools. *Table 2* groups the hyper-parameters that differ from FLAX and OPTAX default settings ; everything else (such as weight initialiser methods, adam learning rate, . . . ) keeps its library default.

## C.1 Network architecture

If not otherwise stated, all our networks (state encoder, quasimetric embedder, odd detector, actor) are MLPs following the same structure : 3 hidden layers of size 512, GELU layer activation. **Encoder** $\phi$ maps a state $s \in \mathcal{S}$ into a latent space $\mathbb{R}^{16}$, and uses a MLP with residual connections. **Quasimetric** $d$ is an Interval Quasimetric Embedding (IQE) (66) with 64 components, each of size 8. **OOD head** $\psi$ is a scalar-output MLP with layer norm and sigmoid activation. **Actor** uses 0.1 dropout layers. It receives observations $s \in \mathcal{S}$ concatenated with target latent keypoints $z \in \mathbb{R}^{16}$ ; the outputs are a mean action $\mu(a|s, z)$ and a state–independent log-std for each action dimension.

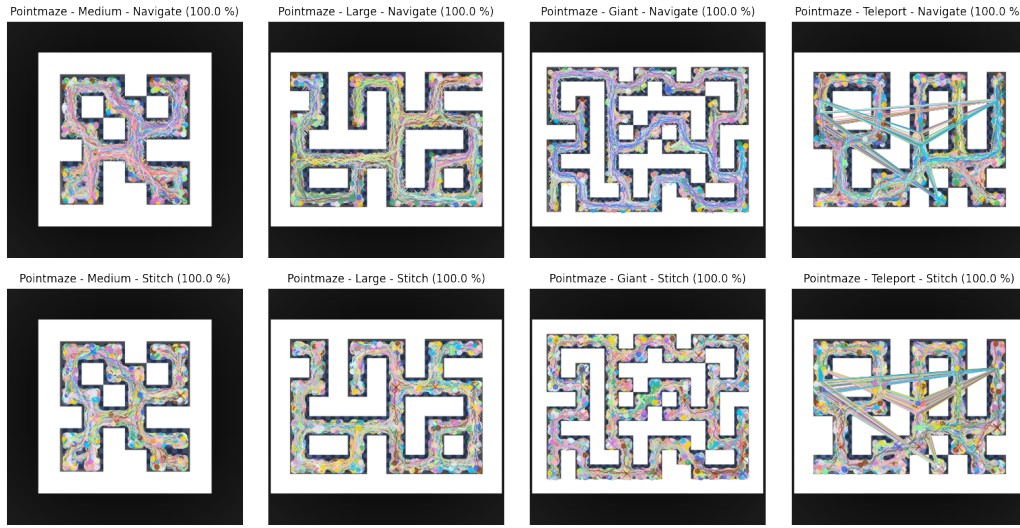

Figure 7: **Illustration of the trajectories within the `navigate` and `stitch` datasets.**

Table 2: Non-default network hyper-parameters used in all experiments.

| MLP | Hidden Sizes | Dropout | Residual | Layer Norm | Output Dim. | Output Act. |
|------|----------------|---------|----------|------------|-------------|-------------|
| $\phi$ | [ 512 , 512 , 512 ] | 0.0 | ✓ | ✗ | 16 | Identity |
| $d$ | [ 512 , 512 , 512 ] | 0.0 | ✗ | ✗ | 512 | Identity |
| $\psi$ | [ 512 , 512 , 512 ] | 0.0 | ✓ | ✓ | 1 | Sigmoid |
| $\pi$ | [ 512 , 512 , 512 ] | 0.1 | ✓ | ✗ | 2 | Tanh mean |

## C.2  Optimisation

All modules use an ADAM optimizer with a learning rate $\eta = 3 \times 10^{-4}$, and a batch size $B = 1024$. Experiments run for $10^6$ gradient steps, on three seeds $(100, 200, 300)$ per maze and dataset. In average each experiment took between $6$ and $8$ hours, on a single Nvidia GeForce GTX 1080 GPU.

We set $\lambda_{\text{OOD}}$ and $\lambda_{\text{dist}}$ as trainable scalar parameters (`Parameter` modules) and update them through the quadratic penalty of their respective loss ; they quickly settle around $10^3$, obviating manual tuning. We also consider IQE margin as $\epsilon = 0.25$, softplus scale as $0.01$, offset as $500$, and a distance cap $\tau = 100$. Moreover the Advantage Weighted Regression (52) temperature is set to $\alpha = 5.0$.

## C.3  Keypoints

We use $K = 100$ keypoints. They were initialized as random latent state from the dataset considered, and remain frozen during the first $10^5$ updates to let $\phi, d, \psi$ stabilise. Thereafter they are optimised with the composite loss $\mathcal{L}_{\text{kps}}$, with a repel strength set to $100$ and a repel range set $100$.

## C.4  Data pipeline

At each gradient step, batches are sampled i.i.d. from the $1$ M-transition replay buffer considered. Goals for the value loss computation are drawn uniformly from the dataset ; while goals for the actor loss follow HER (2) trajectory-based geometric sampling, with a $\lambda = 0.99$ parameter.

## D Learning a State-Space Mapping

### D.1 Learning an Out-of-Distribution Classifier

#### D.1.1 Theoretical Analysis

In this appendix, we give a self-contained analysis showing that, under reasonable assumptions, the learned out-of-distribution function successfully separates reachable states from unreachable ones.

As a reminder, to keep the learned keypoints inside reachable areas, we train an encore $\phi : \mathbb{R}^n \to \mathbb{R}^d$, and an OOD classifier $\psi : \mathbb{R}^d \to [0, 1]$. The difficulty is that the considered dataset $\mathcal{D}$ only supplies *positive* samples (in-distribution). We therefore generate *negative* samples (out-of-distribution) under the assumption that : (i) the state space $\mathcal{S} \subset \mathbb{R}^n$ is compact for the Euclidean topology ; (ii) generated states, noted $\widetilde{\mathcal{D}}$, cover any point at most at a Euclidean distance $Diam(\mathcal{D})$ from $\mathcal{D}$.

In our learning framework, any generated state should be unreachable according to the classifier $\psi$, unless it matches a positive state (*Section 4.2*). Using these generated states, we learn $\psi$ with a binary cross-entropy loss with a dynamic Lagrange multiplier to enforce high confidence on positives while pushing negatives in-distribution probability towards 0 :

- *Positive* samples : $z = \phi(s)$, $s \in \mathcal{D}$, with the target $\psi(z) = 1$ ;
- *Negative* samples : $\tilde{z} = \phi(\tilde{s})$, $\tilde{s} \in \widetilde{\mathcal{D}}$, with the target $\psi(\tilde{z}) = 0$ ;
- **Loss :** $\mathcal{L}_{\text{ood}}(z, \tilde{z}) = -\lambda_{\text{ood}} \cdot \log\big(\psi(z)\big) - \log\big(1 - \psi(\tilde{z})\big)$ ,

where $\lambda_{\text{ood}}$ is updated until satisfaction of a penalty $\Big((1-\delta) - \psi(z)\Big)^2$, with $0 < \delta < 1$ the compliance.

Throughout this analysis we consider (see *Appendix D.1.2* for justifications) :

– A compact state space $\mathcal{S} \subset \mathbb{R}^n$ for the Euclidean topology : it is the set of all in-distribution states.

– The dataset $\mathcal{D} \subset \mathcal{S}$, satisfies an $\epsilon$-density condition : $\exists \, \epsilon > 0, \forall \, s \in \mathcal{S}, \exists \, s' \in \mathcal{D} : \|s - s'\|_2 \leq \epsilon$ .

– We note as $Diam(\mathcal{D})$ the diameter of the dataset $\mathcal{D} : Diam(\mathcal{D}) = \max_{s,s' \in \mathcal{D}} \|s - s'\|_2$ .

– The set of generated samples $\widetilde{\mathcal{D}}$ is $\epsilon$-dense in a *shell* around the dataset $\mathcal{D}$, i.e. :

$$\forall \, s \in \mathbb{R}^n \, , \, \min_{s' \in \mathcal{D}} \|s - s'\|_2 \leq Diam(\mathcal{D}) \Rightarrow \exists \, \tilde{s} \in \widetilde{\mathcal{D}} : \|s - \tilde{s}\|_2 \leq \epsilon \, . \tag{12}$$

– The encoder network $\phi : \mathbb{R}^n \to \mathbb{R}^d$, is assumed $L_\phi$-Lipschitz (65) :

$$\forall \, s, s' \in \mathbb{R}^n : \|\phi(s) - \phi(s')\|_2 \leq L_\phi \cdot \|s - s'\|_2 \, . \tag{13}$$

– The ood classifier network $\psi : \mathbb{R}^d \to [0, 1]$, assumed $L_\psi$-Lipschitz (65) :

$$\forall \, z, z' \in \mathbb{R}^d : |\psi(z) - \phi(z')| \leq L_\psi \cdot \|z - z'\|_2 \, . \tag{14}$$

– After convergence of $\psi$, given the compliance $0 < \delta < 1$ , we have (*Section 4.2*) :

$$\forall \, s \in \mathcal{D} : \psi(\phi(s)) \geq 1 - \delta \, . \tag{15}$$

– After convergence of $\psi$, given a control margin $0 < \eta$ , we note :

$$\bar{\delta} = \max \big\{ \, \psi(\phi(\tilde{s})) \, \big| \, \tilde{s} \in \widetilde{\mathcal{D}} \, , \, \eta \leq \min_{s \in \mathcal{D}} \|\tilde{s} - s\|_2 \, \big\} \, . \tag{16}$$

Concretely, in the assumptions above, the value $\bar{\delta}$ is the worst-case prediction of $\psi$ over all generated points (through interpolation or extrapolation) at least $\eta$ away from any state within the dataset $\mathcal{D}$.

**Lemma D.1.** *In-Distribution Generalization Guarantee :*

*For any $s \in \mathcal{S} : \psi(\phi(s)) \geq 1 - \delta - (L_\psi L_\phi)\epsilon$ .*

*Proof.* Let's consider $s \in \mathcal{S}$. Since the dataset $\mathcal{D}$ is $\epsilon$-dense in $\mathcal{S}$, we know $\exists \, s' \in \mathcal{D}, \ \|s - s'\|_2 \leq \epsilon$, and by Lipschitz continuity :

$$|\psi(\phi(s)) - \psi(\phi(s'))| \leq L_\psi \cdot \|\phi(s) - \phi(s')\|_2 \leq (L_\psi L_\phi) \cdot \|s - s'\|_2 \leq (L_\psi L_\phi)\epsilon$$

In particular : $\psi(\phi(s)) - \psi(\phi(s')) \geq -(L_\psi L_\phi)\epsilon$ . By considering this inequality and *Equation 15* :

$$\psi(\phi(s)) \geq \psi(\phi(s')) - (L_\psi L_\phi)\epsilon \geq 1 - \delta - (L_\psi L_\phi)\epsilon \ .$$

This conclude our proof, and demonstrates the near in-distribution guarantee for reachable states. $\square$

**Lemma D.2.** *Out-of-Distribution Generalization Guarantee :*

*For any $s \in \mathbb{R}^n : \eta \leq \min_{s' \in \mathcal{D}} \|s - s'\|_2 \leq Diam(\mathcal{D}) \ \Rightarrow \ \psi(\phi(s)) \leq \bar{\delta} + (L_\psi L_\phi)\epsilon$ .*

*Proof.* Let's consider $s \in \mathbb{R}^n$ with $\rho = min_{s' \in \mathcal{D}} \|s - s'\|$ that verifies $\eta \leq \rho \leq Diam(\mathcal{D})$. Since the set of generated states $\widetilde{\mathcal{D}}$ is $\epsilon$-dense in a shell around $\mathcal{D}$, up to a distance $Diam(\mathcal{D}$, we know $\exists \, s' \in \widetilde{\mathcal{D}}, \ \|s - s'\|_2 \leq \epsilon$, and by Lipschitz continuity :

$$|\psi(\phi(s)) - \psi(\phi(s'))| \leq L_\psi \cdot \|\phi(s) - \phi(s')\|_2 \leq (L_\psi L_\phi) \cdot \|s - s'\|_2 \leq (L_\psi L_\phi)\epsilon$$

In particular : $\psi(\phi(s)) - \psi(\phi(s')) \leq (L_\psi L_\phi)\epsilon$ . By considering this inequality and *Equation 16* :

$$\psi(\phi(s)) \leq \psi(\phi(s')) + (L_\psi L_\phi)\epsilon \leq \bar{\delta} + (L_\psi L_\phi)\epsilon \ .$$

This demonstrates in and out-of-distribution guarantees within a controlled shell (*Theorem 4.1*). $\square$

### D.1.2 Discussion of Standing Assumptions

We now expand on why the hypothesis in *Appendix D.1.1* are both natural and reasonably attainable.

**Compact State Space $\mathcal{S} \subset \mathbb{R}^n$ :** To the best of our knowledge, most navigation tasks in the literature, and all in OGBench (49), can be restricted to a bounded subset of $\mathbb{R}^n$. Enforcing that $\mathcal{S}$ is closed and bounded, and consequently compact in finite dimension, simply means there are well-defined "walls" or "obstacles" beyond which the agent never ventures. In the majority of simulated mazes, for example, the reachable region is a finite polygonal domain.

**Dataset $\epsilon$-Density in $\mathcal{S}$ :** In Offline RL, one can typically records many (possibly millions of) trajectories covering the state space (see *Appendix B.2*). It is therefore reasonable to assume that for every reachable point $s \in \mathcal{S}$, there is some state $s' \in \mathcal{D}$ within a small Euclidean radius $\epsilon$. In practice, this radius can be made arbitrarily small by collecting more trajectories or, in maze navigation, by performing stratified sampling along the map's corridor.

**Dataset Diameter $Diam(\mathcal{D})$ :** We define $Diam(\mathcal{D}) = \min_{s, s' \in \mathcal{D}} \|s - s'\|_2$. This is simply the farthest distance between any two dataset points. In a bounded maze, $Diam(\mathcal{D})$ is finite. It is used to cap how far we need to sample negatives : beyond $Diam(\mathcal{D})$, we consider everything is out of reach.

**Negative Sampling with $\widetilde{\mathcal{D}}$ :** We assume that our negative sampler can produce, for every point $s \in \mathbb{R}^n$ whose distance to $\mathcal{D}$ is at most $Diam(\mathcal{D})$, at least one generated sample $\tilde{s}$ within $\epsilon$. In other words, the set $\widetilde{\mathcal{D}}$ is $\epsilon$-dense in the controlled shell $\{ \, s \mid s \in \mathbb{R}^n \, , \, \min_{s' \in \mathcal{D}} \|s - s'\|_2 \leq Diam(\mathcal{D}) \, \}$. Concretely, one can implement this by drawing uniform random points in the ball of radius $Diam(\mathcal{D})$ around each $s \in \mathcal{D}$. In low dimension ($n = 2$ or $n = 3$), one can use a grid to guarantee $\epsilon$-covering.

**Lipschitz Continuity :** MLPs with Lipschitz activations (such as ReLU, GELU, or even Sigmoid), and eventually layer normalization, are provably Lipschitz (65). For example in a MLP with K hidden layers, if every weight matrix $W_i$ has $\|W_i\| \leq \kappa$ and each activation is 1-Lipschitz, then the MLP itself is L-Lipschitz, with $L = \kappa^K$. In practice, one can enforce weight clipping, weight regularization, spectral-norm regularization, or to keep the Lipschitz coefficient relatively small (43).

**Positive Samples Constraint :** During training of the classifier $\psi$, each positive state $s \in \mathcal{D}$ incurs a loss $-\lambda_{\text{ood}} \cdot \log\left(\psi(\phi(s))\right)$ with $\lambda_{\text{ood}}$ steadily increased until $\psi(\phi(s)) \geq 1 - \delta$ for all $s$. That is a simple, yet effective, Lagrangian strategy to force $\psi$ into $[1 - \delta, 1]$ for positive samples.

**Negative Samples Worst-Case :** By construction, any $\tilde{s} \in \widetilde{\mathcal{D}}$ incurs loss $-\log\left(1 - \psi(\phi(\tilde{s}))\right)$ which pushes $\psi(\phi(\tilde{s}))$ toward 0. We define $\bar{\delta} = \max\{\ \psi(\phi(\tilde{s}))\ \big|\ \tilde{s} \in \widetilde{\mathcal{D}}\ ,\ \eta \leq \min_{s \in \mathcal{D}} \|\tilde{s} - s\|_2\}$. In practice, one cannot guarantee $\bar{\delta} = 0$ exactly, but empirical training drives $\bar{\delta}$ to a small value.

Because each assumption can be well approximated in navigation settings, the *Lemmas D.1* and *D.2* become meaningful guarantees : with $\epsilon$, $\delta$, and $\bar{\delta}$ small enough, $\psi$ *carves out* the reachable regions.

### D.1.3 Discussion of Limitations

While the above analysis is mathematically sound, several practical limitations deserve attention.

**High-Dimensional State Spaces (large n) :** In $\mathbb{R}^n$ with $n > 2$, uniformly covering the previously defined shell up to a fine $\epsilon$ becomes exponentially costly in $n$. Grid-based sampling will require $O(\epsilon^{-n})$ points. In high-dimensional pixel-based observations (e.g. $n \simeq 1000$), one cannot hope to sample negatives points in that entire ball. Instead we should exploit structures such as known game map geometry, or pre-trained world models to propose out-of-distribution samples more intelligently.

**Interpolation & Extrapolation vs. Uniform Sampling :** Our proposed `ProQ` implementation use one-step linear interpolation $(1 - \alpha) \cdot s_1 + \alpha \cdot s_2$ and extrapolation $(1 + \beta) \cdot s_1 - \beta \cdot s_2$, with $\alpha, \beta \sim \mathcal{U}([0, 1])$ and $s_1, s_2 \in \mathcal{D}$. These do not uniformly cover the entire negative shell in $\mathbb{R}^n$ for $n > 2$, they only explore the convex hull of pairs and its immediate outward extensions. Hence, there may exist unreachable points $s$ within the admissible shell that never get generated, leaving $\psi$ unconstrained. A uniform-shell sampler would ensure no such gap, but is expensive when $n$ is large.

### D.2 Coulomb-like Energy & Uniform Keypoint Spreading

In this appendix, we discuss and analyze why minimizing a Coulomb-like repulsive energy over a finite set of learnable keypoints $\{p_i\}_{i=1}^K \subset \mathbb{R}^d$ pushes those keypoints to spread *uniformly* over a compact domain $\phi(\mathcal{S})$. We begin by recalling the classical Coulomb law in physics, then show how our loss in `ProQ` reduces to a pure repulsive energy in latent space. Finally, we prove under mild assumptions that any minimizer of this energy must maximize pairwise separation and, asymptotically, approaches an uniform distribution.

### D.2.1 From Coulomb's Law to Our Energy Loss

In electrostatics, two positively-charged particles located at $x_i, x_j \in \mathbb{R}^3$ experience a potential :

$$V(x_i, x_j) = \frac{q_i q_j}{4\pi\epsilon_0} \cdot \frac{1}{\|x_i - x_j\|_2} \tag{17}$$

where $q_i, q_j > 0$ are their charges, $\epsilon_0$ is the vacuum permittivity, and $\|\cdot\|_2$ is the Euclidean norm.

For K charges, when all charges are equal ($q_i = q_j$), we often only study the total potential energy :

$$E(x_1, \ldots, x_K) = \sum_{1 \leq i < j \leq K} \frac{1}{\|x_i - x_j\|_2} \ . \tag{18}$$

Minimizing this energy $E(x_1, \ldots, x_K)$ on a compact domain (e.g. the unit sphere $\mathbb{S}^2$) yields the classical Thomson Problem (7; 33) of finding well-separated, nearly-uniform point configurations.

In `ProQ`, we replace Euclidean distance by a learned latent *quasimetric* $d : \mathbb{R}^d \times \mathbb{R}^d \to \mathbb{R}^+$. Concretely, if $\{z_k\}_{k=1}^K$ are the embeddings of the latent learnable keypoints, then our Coulomb-like repulsive loss becomes :

$$\mathcal{L}_{\text{repel}} = \sum_{i \neq j} \frac{1}{d(z_i, z_j) + \epsilon} \,, \tag{19}$$

where $\epsilon > 0$ is a small constant that prevents numerical blow-up when two points become very close.

Repulsion alone could push keypoints outside the region covered by the dataset. We therefore add a soft barrier derived from the OOD classifier $\psi$, penalising points that leave the in-distribution set :

$$\mathcal{L}_{\text{ood}} = -\lambda_{\text{ood}} \cdot \sum_{k=1}^K \log \psi(z_k), \tag{20}$$

with $\lambda_{\text{ood}}$ updated until satisfaction of a penalty $\left((1 - \delta) - \sum_{k=1}^K \psi(z_k)\right)^2$, with $0 < \delta < 1$ the compliance. In other words, on average, all latent keypoints must stay in-distribution to some extent.

The energy-based losses add up to $\mathcal{L}_{\text{kps}} = \mathcal{L}_{\text{rep}} + \mathcal{L}_{\text{ood\_kps}}$. At each gradient step, while optimizing this composed loss, $\phi$, $d$, and $\psi$ are frozen to avoid instabilities in the learning of the latent space.

### D.2.2 Theoretical Analysis

In this section, we analyze the geometric properties of a set of $K$ keypoints $\{s_k\}_{k=1}^K$ within a compact state space $\mathcal{S} \subset \mathbb{R}^n$, for the Euclidean topology, that minimize a repulsive energy function $E : \mathcal{S}^K \to \mathbb{R}^+ \cup \{+\infty\}$ based on a continuous quasimetric $d : \mathcal{S} \times \mathcal{S} \to \mathbb{R}^+$. The goal is to understand how effectively such a function can lead to keypoints that provide a good coverage of $\mathcal{S}$.

This analysis considers an idealized scenario where the learned keypoints have achieved a global minimum of the repulsive energy, abstracting away from the iterative optimization process and the influence of the OOD barrier while learning the keypoints (which keeps them in-distribution areas).

Throughout this analysis we consider :

– A compact state space $\mathcal{S} \subset \mathbb{R}^n$ for the Euclidean topology, not reduced to a single point if $K \in \mathbb{N}^*$.

– A continuous quasimetric $d : \mathcal{S} \times \mathcal{S} \to \mathbb{R}^+$ uniformly equivalent to the Euclidean distance :

$$\exists\, c_1, c_2 > 0, \; \forall x, y \in \mathcal{S}, \; c_1 \cdot \|x - y\|_2 \leq d(x, y) \leq c_2 \cdot \|x - y\|_2 \tag{21}$$

– An energy function $E : \mathcal{S}^K \to \mathbb{R}^+ \cup \{+\infty\}$ defined as :

$$E : \mathcal{S}^K \to \mathbb{R}^+ \cup \{+\infty\} \,, \; s_1, \ldots, s_K \to \sum_{i \neq j} \frac{1}{d(s_i, s_j)} \tag{22}$$

– A metric $d' : \mathcal{S} \times \mathcal{S} \to \mathbb{R}^+$ defined as : $\forall x, y \in \mathcal{S}, \; d'(x, y) = \min\big(d(x, y), d(y, x)\big)$ .

**Lemma D.3.** *Compactness of* $(\mathcal{S}, d')$ *and Equivalence of Topologies :* $(\mathcal{S}, d')$ *forms a compact metric space. The topology induced by $d'$ on $\mathcal{S}$ is identical to the Euclidean topology on $\mathcal{S}$.*

*Proof.* Given the assumptions (*Equation 21*), for any $x, y \in \mathcal{S}$ : $d'(x, y) \leq d(x, y) \leq c_2 \cdot \|x - y\|_2$ and as : $c_1 \cdot \|x - y\|_2 \leq d(x, y)$ , $c_1 \cdot \|y - x\|_2 \leq d(x, y)$ , and $\|x - y\|_2 = \|y - x\|_2$ we have :

$$c_1 \cdot \|x - y\|_2 \leq d'(x, y) \leq c_2 \cdot \|x - y\|_2 \tag{23}$$

Therefore $d'$ is uniformly equivalent to the Euclidean distance, which implies topological equivalence. Thus, since $\mathcal{S}$ is compact in the Euclidean topology, it is also compact in the one induced by $d'$.  $\square$

**Theorem D.4.** *Existence and Distinctness of Energy Minimizers : Let's consider the energy function* $E : \mathcal{S}^K \to \mathbb{R}^+ \cup \{+\infty\}$ *. Under the previous assumptions of compactness and continuity we have :*

*(i) E has a finite minimum, and there exists a configuration* $\{s_k^*\}_{k=1}^K \subset \mathcal{S}$ *that reaches it.*
*(ii) For any such minimizer* $\{s_k^*\}_{k=1}^K \subset \mathcal{S}$ *, if* $K > 1$ *then all keypoints are distinct.*

*Proof.* If $\mathcal{S} \subset \mathbb{R}^n$ is reduced to a single point, the proof is direct (the energy is $0$ as the empty sum).

If $\mathcal{S}$ is not reduced to a single point, we can select a configuration $\{u_k\}_{k=1}^K \subset \mathcal{S}$ of $K$ distinct points. Then $\forall\, i \neq j$ , $u_i \neq u_j$, we have $\forall\, i \neq j$ , $d(u_i, u_j) \neq 0$ and $M_o = E(u_1, \ldots, u_K) > 0$ is finite.

We consider the set $\mathcal{C} = \{\{s_k\}_{k=1}^K \in \mathcal{S}^K \mid E(s_1, \ldots, s_K) \leq M_0\}$.

If $\{s_k\}_{k=1}^K \in \mathcal{C}$ then all its points are distinct, as otherwise there exist $s_i = s_j$ and $d(s_i, s_j) = 0$ which would lead to $E(s_1, \ldots, s_K) = +\infty$ contradicting $E(s_1, \ldots, s_K) \leq M_0$. In particular :

$$\forall\, i \neq j\,, \ \frac{1}{d(s_i, s_j)} \leq \sum_{i' \neq j'} \frac{1}{d(s_{i'}, s_{j'})} \leq M_0 \tag{24}$$

Which leads to :

$$\forall\, \{s_k\}_{k=1}^K \in \mathcal{C}\,, \ \forall\, i \neq j\,, \ d(s_i, s_j) \geq \frac{1}{M_0} \tag{25}$$

Let's consider $\big(\{s_k^{(n)}\}_{k=1}^K\big)_{n\in\mathbb{N}}$ be a converging sequence in $\mathcal{C}$ with a limit $\{s_k^*\}_{k=1}^K \in \mathcal{S}^K$.

By continuity of $d$ we have :

$$\forall\, i \neq j\,, \ d(s_i^{(n)}, s_j^{(n)}) \geq \frac{1}{M_0} \Rightarrow \lim_{n \to +\infty} d(s_i^{(n)}, s_j^{(n)}) = d(s_i^*, s_j^*) \geq \frac{1}{M_0}$$

As $d$ is continuous, each term $d(s_i, s_j)^{-1}$ is a continuous function of $s_i$ and $s_j$ in a neighborhood of $(s_i^*, s_j^*)$ where the denominator is non-zero. Thus $E$, as a finite sum, is continuous at $\{s_k^*\}_{k=1}^K$ and :

$$E(s_1^{(n)}, \ldots, s_K^{(n)}) \leq M_0 \Rightarrow \lim_{n \to +\infty} E(s_1^{(n)}, \ldots, s_K^{(n)}) = E(s_1^*, \ldots, s_K^*) \leq M_0$$

Thus $\mathcal{C}$ contains all its limit points and therefore is closed. Since $\mathcal{S}^K$ is compact, as a product of compact set, $\mathcal{C}$ is also compact. Moreover, the function $E$ is continuous on the non-empty compact set $\mathcal{C}$ (as shown, all points in $\mathcal{C}$ have bounded distance away from zero, ensuring continuity of $E$). By the extreme value theorem, $E$ attains its minimum on $\mathcal{C}$ at some configuration $\{s_k^*\}_{k=1}^K$. This minimum value $E(s_1^*, \ldots, s_K^*)$ is finite and at most $M_0$ and is the global minimum over $\mathcal{S}^K$. Indeed, any configurations $\{u_k\}_{k=1}^K$ outside $\mathcal{C}$ either have $E(u_1, \ldots, u_K) > M_0$ or $E(u_1, \ldots, u_K) = +\infty$. Hence, recalling *Equation 25*, it comes that every points of the minimizer $\{s_k^*\}_{k=1}^K$ are distinct. $\quad\square$

### D.3 Learned State-Space Mappings

*Figure 8* shows the learned state mappings. In the Medium maze, corridors are consistently identified as in-distribution, and the red keypoints uniformly occupy each hallway. We notably notice that varying the seed does not significantly alter the learned latent coverage. Similarly, in the Large maze, the OOD boundary tightly traces the walls, and keypoints remain evenly spaced along identical corridors for all seeds, confirming robustness. In the Teleport environment, the portal squares introduce stochastic transitions ; the OOD classifier occasionally colors portal cells green (intermediate confidence), and keypoints cluster variably near these portals for different seeds, indicating sensitivity to random initialization in regions with non-deterministic dynamics. Finally, in the Giant maze, the expansive corridor network is consistently recognized as in-distribution, and keypoints again spread uniformly across all three seeds, illustrating that ProQ's mapping remains stable even at larger scale.

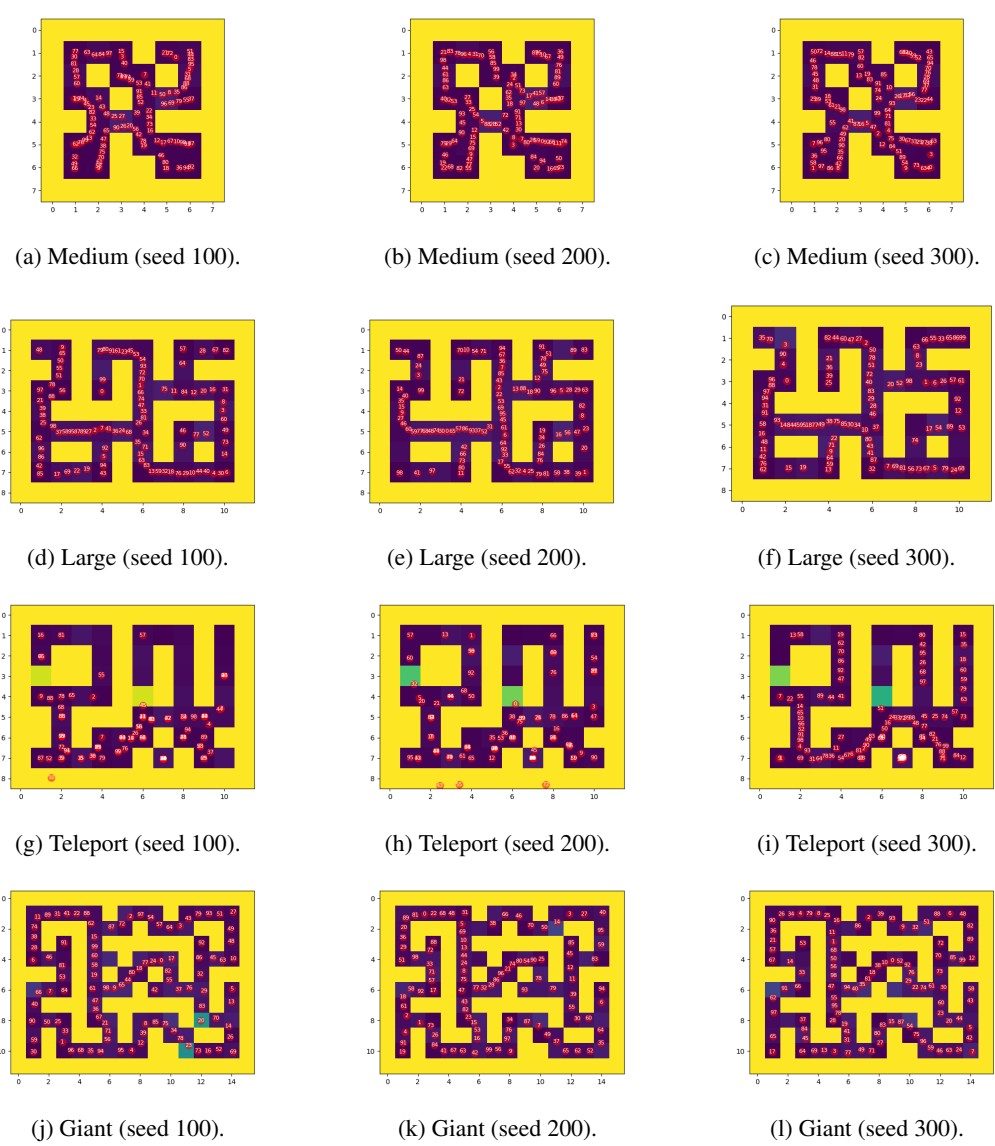

(a) Medium (seed 100).    (b) Medium (seed 200).    (c) Medium (seed 300).

(d) Large (seed 100).    (e) Large (seed 200).    (f) Large (seed 300).

(g) Teleport (seed 100).    (h) Teleport (seed 200).    (i) Teleport (seed 300).

(j) Giant (seed 100).    (k) Giant (seed 200).    (l) Giant (seed 300).

Figure 8: **Learned State-Space Mappings.** Each subplot displays the OOD classifier's output (from *yellow* for high OOD probability, to *purple* for high in-distribution probability) along with the learned keypoints as red dots. Each of the rows correspond to a different environment (Medium, Large, Teleport, and Giant), and each of the columns correspond to a different seeding (100, 200, and 300).

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
