# OpenReview forum: "Offline Goal-Conditioned Reinforcement Learning with Projective Quasimetric Planning"
_rl-conference.cc/RLC/2025/Workshop/RLVG — RLVG Workshop - RLC 2025_

### Official Review · Reviewer_LK2p · 2025-06-13
**Review of Paper: Offline Goal-Conditioned Reinforcement Learning with Projective Quasimetric Planning**

**Rating:** 3
**Confidence:** 4

**Summary:**

The authors propose Projective Quasimetric Planning (ProQ), a fully-offline, goal-conditioned RL framework. ProQ jointly learns (i) a quasimetric representation of action reachability, (ii) an OOD classifier that defines the reachable manifold of states, and (iii) a set of Coulomb-repelled latent keypoints that sparsely cover that manifold. Greedy hops over the keypoint graph plus a short-horizon AWR controller achieve state-of-the-art success rates on long-horizon PointMaze navigation tasks, outperforming six recent baselines.

**Strengths:**

Strengths:
* Clever composition – unifies metric learning, coverage, and OOD detection into one latent space.
* Memory-efficient – fixed number of keypoints; unlike nearest-neighbour planners memory does not grow with dataset size.
* Empirical results – large performance gap on giant maze.
* Ablation study clearly shows OOD barrier is necessary.
* Paper is mostly well written; figures nicely illustrate latent geometry.

**Weaknesses:**

ProQ’s geometric idea is promising, but the authors should extend the benchmark suite, dig into stochastic failure modes, quantify coverage and memory, and tighten theory/robustness comparisons to fully substantiate the method.
* Evaluation is limited to PointMaze; no manipulation, vision, or game-engine benchmarks – external validity unclear.
* Stochastic “Teleport” variant exposes fragility (43 – 35 % success); root cause not analysed.
* Comparison set lacks strong model-based offline planners (e.g. MOPO, Dreamer-V3).

**Best Paper Nomination:**

No

**Claims:**

The paper does a solid job substantiating its core performance claim—that ProQ outperforms strong offline RL baselines on long-horizon navigation—through a thorough PointMaze study and an insightful OOD ablation. However, claims about scalability, uniform coverage, generality to manipulation, and theoretical guarantees are under-evidenced.

**Suggestions:**

The paper would certainly benefit from an extended experimental setup beyond point navigation.

---

### Official Review · Reviewer_UU4u · 2025-06-15
**Review for Projective Quasimetric Planning**

**Rating:** 3
**Confidence:** 3

**Summary:**

The authors present Projective Quasimetric Planning (ProQ), an algorithm for planning in offline goal-conditioned RL settings that is inspired by quasimetrics and physics. The authors benchmark ProQ in several maze-solving environments, and these experiments show that ProQ performs well relative to competitor algorithms.


I am having trouble deciding between accept and reject. On the one hand, the combination of physics-inspired methods and RL is interesting, and the results provided by the algorithm are strong. On the other hand, the paper's narrative is unfocused, which significantly harms understanding. However, because the submission venue is a workshop, I believe the authors and their work could benefit greatly from conversion with others. I therefore recommend accepting.

**Strengths:**

The algorithm introduced in the paper appears to provide strong performance in the domain tested.

I also like the creativity of combining physics and ML. More of this, please!

**Weaknesses:**

The paper's narrative is unclear and unfocused, which significantly harms understanding. Please see the "Suggestions" section for a few examples.

**Best Paper Nomination:**

No

**Claims:**

It is hard to discern the claims in the paper due to the unfocused narrative.

**Suggestions:**

Below are a few notes I took while reading the work:

The writing is a little unfocused. E.g., in paragraphs 2-5 of the Introduction, the authors touch on numerous challenges in many different settings but never clarify which challenge and what setting they are studying and why. I would recommend to narrow the Introduction's focus on the specific problem being studied (long horizon navigation?).

Also, the paper could benefit from an earlier definition of terms. The paragraph starting at line 46 uses terms like quasimetric and OOD discriminator. After some thought, a reader could guess what the authors mean, but the paper would read better if these things were briefly defined beforehand.

Typo line 104: the lefthand bracket is backward in gamma's domain definition.

Line 106: subscript \mathcal{M} is used with the notation for optimal parameters, expected cumulative return, and success rate but is not defined.

In a sentence starting in line 114, the authors state that "optimal value function(s)... often exhibit asymmetry". It would be helpful if the authors would expand on what they mean by "asymmetry", why value functions exhibit this characteristic, and why it might be a problem.

---

### Decision · Program_Chairs · 2025-06-19

**Decision:**

Accept

**Comment:**

This paper introduces **Projective Quasimetric Planning (ProQ)**, an offline goal-conditioned reinforcement learning algorithm inspired by quasimetrics and physics, demonstrating strong performance in maze-solving environments.

The strengths of ProQ include its novel composition of metric learning, coverage, and out-of-distribution (OOD) detection into a single latent space, leading to strong empirical results and memory efficiency.

However, the paper's narrative is not very clear and the evaluation (restricted to maze environments) is limited, lacking benchmarks in more complex settings like manipulation or vision. The paper could be improved with additional analysis of stochastic failure modes or by providing a comprehensive comparison with other strong model-based offline planners. We encourage the authors to address these points, particularly improving the clarity and providing a deeper analysis of the method's robustness, in the camera-ready version to be presented at the workshop.